# Deep-Learning-Based Algorithm for the Removal of Electromagnetic Interference Noise in Photoacoustic Endoscopic Image Processing

**DOI:** 10.3390/s22103961

**Published:** 2022-05-23

**Authors:** Oleksandra Gulenko, Hyunmo Yang, KiSik Kim, Jin Young Youm, Minjae Kim, Yunho Kim, Woonggyu Jung, Joon-Mo Yang

**Affiliations:** 1Center for Photoacoustic Medical Instruments, Department of Biomedical Engineering, Ulsan National Institute of Science and Technology (UNIST), Ulsan 44919, Korea; sasha@unist.ac.kr (O.G.); rltlr06@unist.ac.kr (K.K.); happyjinyoung.youm@gmail.com (J.Y.Y.); kimminjae519@unist.ac.kr (M.K.); 2Translational Biophotonics Lab, Department of Biomedical Engineering, UNIST, Ulsan 44919, Korea; hmyang@unist.ac.kr; 3Department of Mathematical Sciences, UNIST, Ulsan 44919, Korea

**Keywords:** electromagnetic interference noise, noise removal, convolutional neural network, image-to-image regression, deep learning, photoacoustic tomography, photoacoustic microscopy, photoacoustic endoscopy, microvasculature visualization

## Abstract

Despite all the expectations for photoacoustic endoscopy (PAE), there are still several technical issues that must be resolved before the technique can be successfully translated into clinics. Among these, electromagnetic interference (EMI) noise, in addition to the limited signal-to-noise ratio (SNR), have hindered the rapid development of related technologies. Unlike endoscopic ultrasound, in which the SNR can be increased by simply applying a higher pulsing voltage, there is a fundamental limitation in leveraging the SNR of PAE signals because they are mostly determined by the optical pulse energy applied, which must be within the safety limits. Moreover, a typical PAE hardware situation requires a wide separation between the ultrasonic sensor and the amplifier, meaning that it is not easy to build an ideal PAE system that would be unaffected by EMI noise. With the intention of expediting the progress of related research, in this study, we investigated the feasibility of deep-learning-based EMI noise removal involved in PAE image processing. In particular, we selected four fully convolutional neural network architectures, U-Net, Segnet, FCN-16s, and FCN-8s, and observed that a modified U-Net architecture outperformed the other architectures in the EMI noise removal. Classical filter methods were also compared to confirm the superiority of the deep-learning-based approach. Still, it was by the U-Net architecture that we were able to successfully produce a denoised 3D vasculature map that could even depict the mesh-like capillary networks distributed in the wall of a rat colorectum. As the development of a low-cost laser diode or LED-based photoacoustic tomography (PAT) system is now emerging as one of the important topics in PAT, we expect that the presented AI strategy for the removal of EMI noise could be broadly applicable to many areas of PAT, in which the ability to apply a hardware-based prevention method is limited and thus EMI noise appears more prominently due to poor SNR.

## 1. Introduction

Electromagnetic (EM) interference (EMI) noise is one of the important types of noise that disturbs the normal operation of an electrical circuit, and the related issues have been an important subject in many areas of electronics over the past century [1,2,3,4,5,6,7]. This noise is normally caused by external sources through mechanisms such as electromagnetic induction, electrostatic coupling, and conduction. Although the basic hardware platform for most modern electronic devices has evolved for digital-based operations, achieving a sufficiently stable circuit that is unaffected by EMI noise is still seen as a significant issue in many areas of electronics—in particular, in telecommunication technology, which deals with such important devices as radio, television, and cellular phones in the radio frequency (RF) or even higher frequency bands [6]. This is because the transmitted or received signal itself is based on analogue at the lowest level. Although, EMI noise is known to originate from any current or voltage source working in a switching or alternating mode near the circuit of interest, it is quite difficult to avoid such noise because the level and pattern of the disturbance are greatly affected by the geometric factors of the conducting path, by the spatial configuration of the associated elements, or by the electric impedance mismatch. Therefore, a circuit can still be affected by EMI noise even though there is no direct electrical connection to any other adjacent object.

EMI noise has also been an ongoing issue in photoacoustic (PA) tomography (PAT) [8,9,10,11,12,13,14,15,16,17] because PAT relies on analogue ultrasound (US) signals typically ranging in the RF band 1–100 MHz. PAT is a novel tomographic imaging modality based on the photoacoustic effect, which has been drawing increased attention as a range of biomedical applications have shown promising results over the past decade. Typically, a PAT system consists of optical illumination and acoustic detection units as well as additional modules to acquire images by scanning, whether electronically or mechanically. Consequently, in the case of a PAT system, a switching-based circuit module, such as a switching mode power supply, a stepping motor and its driver, or a pulser circuit, could all be potential sources of EMI noise. Of course, the commercial devices listed are manufactured with a regulated EM radiation limit. However, EMI noise can still occur if they are not properly engaged with the entire circuit that is being constructed, and if such issues as grounding and impedance matching between them are not carefully considered. If EMI noise is involved in a PAT image that was acquired on the basis of mechanical scanning, the noise pixels will usually appear in random positions, as rain-like striking patterns in a B-mode image presentation format, because a range of data points in an A-line was serially affected. Due to this feature, it is not difficult to recognize noise-affected pixels based on a visual judgment. However, it is almost impossible to remove them by applying a simple filtering method, such as a band-pass filter, because the spectral width of the delta function-like electric disturbance is very wide, and thus there is a spectral overlap with the detection band of the US transducer employed.

In the case of photoacoustic endoscopy (PAE) or intravascular photoacoustic (IVPA) imaging applications [18,19,20,21,22,23,24,25,26,27,28,29], related issues appear more frequently and seriously because it is a common hardware situation that the US transducer for signal detection and the associated preamplifier for the amplification of the detected signal are placed separately, with one at the distal end of the flexible probe section and the other with the proximal driving unit, at a distance of more than ~1.5 m, as the space allowed for installing a preamplifier circuit at the distal scanning tip is very limited. Moreover, due to the same limited space problem, it is not easy to apply proper shielding material to the connection path (i.e., the flexible probe section), which is located between the two parts mentioned [7]. Consequently, it has been recognized that it is very difficult to implement a perfect hardware setup in PAE or IVPA that is not affected by interference noise at all, especially where the system is embodied at a laboratory level, and the problem related to the involvement of noise has been mentioned frequently in many reports [19,20,21,23,25,29].

In this study, we propose a deep-learning-based EMI noise removal algorithm for use on PA images acquired by a newly constructed PAE system [29]. Although multiple studies have applied artificial intelligence (AI) techniques to PAT, all of them were related to other topics, such as PA image classification [30,31], reverberation removal [32], missing data restoration [33,34], artifact removal [35,36,37], reconstruction assistance [38,39], image segmentation [40,41,42], and resolution enhancement [43,44]; more details on the previous works in this area are provided in Appendix A. To the best of our knowledge, no previous studies have addressed the problem of EMI noise removal as our work does. This is true and also natural in some respects because all the previous studies targeted its use when linked to a PAT system with a large footprint, in which EMI noise is not usually involved in an acquired PAT image because necessary action to avoid EMI noise is already taken at a hardware level by applying sufficient shielding. That is, although in principle, it is not impossible to build a nearly perfect endoscopic hardware system that produces only a negligible amount of EMI noise, it would be extremely costly and very labor-intensive, which would not be realistic in most laboratory environments.

Therefore, recognizing the general hardware limitations of a lab-made PAT miniature probe, we considered it is important to develop a software-based EMI noise removal system to prevent researchers from missing any important anatomical structures that might exist in an acquired image but not be recognized due to the noise. Of course, it might be also thought that, due to the aforementioned commonly intervening characteristic of EMI noise in many circuits, there must be already many studies that developed a similar EMI noise removal algorithm. Interestingly, however, we could not find any articles that dealt with such an EMI noise removal issue in the “rain-like” striking pattern like in our case [45,46,47,48,49,50,51,52,53,54]. Among the various types of noise, the salt-and-pepper noise that has been studied in general digital image processing areas in a planar photographic image format appears to be similar to the EMI noise seen in our case in terms of the randomness in a noise-involved position [45,46,47,48]. However, the salt-and-pepper noise presents no structural information, and its noise pixel values take either the maximum or minimum value of the signal dynamic range [45,46,47,48], meaning that it is fundamentally different from the EMI noise seen in our case. Consequently, there was no identified report that addressed a similar EMI noise removal issue when it comes to the medical image processing area in which tomographic images are usually dealt with. Thus, we attribute the rain-like appearance of EMI noise in an image to the unique character of the PAT and US imaging technology that records related signals by consecutively digitizing arriving acoustic waves over a set time interval (i.e., gated time) and presents the recorded image data in a B-mode format. In the case of conventional PAT research, it would be somewhat natural to understand why there were no related studies. We think it was because EMI noise can be sufficiently prevented by a hardware-based treatment, unlike the typical hardware situation faced in PAE or IVPA imaging applications, which emerged relatively later. Thus, although the presented deep-learning-based EMI noise removal strategy would benefit PAE or IVPA research only at this moment, it was our basic idea that related approaches would become increasingly important in the future because the development of low-cost PAT systems or PA sensors is emerging as a new important subject in related areas [13,55]. Moreover, in terms of the morphological features of EMI noise appearing in an image, the topic of this study may have a close linkage to the rain removal problem that occurs in surveillance cameras in relation to security or safety [56,57,58].

To remove EMI noise from a PAE image, simple methods, such as cross-correlation [23] or a transverse signal gradient-based noise detection algorithm [29], have been applied as self-rescue methods in previous studies. We guess that the first method might have worked satisfactorily because the related endoscopic probe was operated in an acoustic-resolution (AR) PAE mode, in which it is typical that PA signals from capillaries are hardly captured and resolved. However, when we applied the latter method to our endoscopic images, the result was not satisfactory, especially when processing signals relating to a fine structure, such as a capillary network, because the non-AI-based, classical deterministic algorithm could not accurately distinguish EMI noise-affected pixels from normal delicate capillary signals resolved by the optical-resolution (OR) PAE. Consequently, the denoising work required a great deal of time and labor for manual-based image segmentation in order to remove the EMI noise that occurred only in the non-intestine area while avoiding the alteration or loss of important capillary signals. Thus, it is our point that, if the operation mode of the PAE or IVPA probe approaches such an optical microscopy level of high resolution, related EMI noise removal becomes tricky, regardless of the type of the applied methods [23,29], because the noise pattern itself becomes similar to that of the capillary signal resolved by the OR PAE; note that, among previous PAE or IVPA studies, in which developing a narrow diameter imaging probe was the key, only three reports demonstrated an OR imaging mode [21,24,29].

Recent studies in the biomedical imaging community have shown the applicability of deep-learning techniques to solve such tedious and complex image-to-image regression problems in super-resolution [59,60], segmentation [40,41,42,61,62], and denoising [63,64] from a variety of imaging modalities. The type of deep neural network commonly used for handling image-to-image regression is a convolutional neural network (CNN) with convolutional layers end-to-end. Among CNNs, fully convolutional neural networks (FCNs) have many benefits in terms of dealing with variable input sizes, conserving the dimensions of the image data and efficient learning from shared weights. Moreover, it can also be trained to extract numerous important features for a given purpose without human supervision [52]. In particular, FCNs are used for semantic segmentation where each image pixel is assigned to one-pixel class. In other words, since FCNs have the ability of pixel-wise classification and modification, FCNs should not only be able to capture the random locations of the EMI noise but also to remove the amount of noise at those pixels. This should also become manifest in comparison with classical computational methods. 

In this article, to deal with EMI noise-affected PAE images more effectively, we consider CNN-based noise removal algorithms built upon four representatives of the fully CNN architectures, in combination with an image-to-image regression technique. Based on the comparison, we propose a CNN-based noise removal algorithm that best achieves our goal and applies it to in vivo data to confirm the suitability of the method for EMI noise removal. Thus, the main contribution of this work is to develop and apply CNN-based algorithms for the first time to remove EMI noise from images acquired from a PAE system. In particular, we consider U-Net, Segnet, FCN-16s, and FCN-8s architectures. Not only do we compare these CNN-based algorithms that are modified to work for the EMI noise removal, but we also consider classical computational algorithms to confirm the superiority of CNN-based ones. At last, we will conclude that the U-Net architecture is the most efficient and accurate among the candidates. Not to mention, the U-Net can generate a denoised 3D vasculature map showing a clear image of the mesh-like capillary networks distributed in the wall of a rat colorectum.

## 2. Materials and Methods

### 2.1. Data source

The dataset utilized for this study was acquired from the colorectums of Sprague Dawley rats (~400 g) and the urinary tract of a New Zealand white rabbit (~1.5 kg) over three independent experiments (colorectum imaging for two rats and transurethral imaging for one rabbit) by using the integrated OR-PAE and endoscopic US imaging system that we recently reported [29]. Figure 1a depicts the approximate experimental setup. The system allows for the acquisition of OR-PAE B-scan images at a frame rate of 20 Hz, based on the 532 nm optical excitation (pulse width: ~2 ns) and subsequent US signal detection, achieved by symmetrically placed dual US transducers (center frequency: 40 MHz, fractional bandwidth: ~60%, physical dimension: 0.6 mm × 0.5 mm × 0.2 mm). The system could also acquire co-registered US images, simultaneously, and the US images were also affected by EMI noise. In this study, however, we did not consider EMI noise removal in US images because it is relatively easy to deal with in comparison with PAE images since the spatial resolution of the US imaging mode falls within the AR level rather than OR.

Considering the 40 MHz center frequency of the employed dual transducers, we recorded OR-PAE B-scan images at a sampling rate of 200 MHz and set the data length of each A-line to be 400 points and the total number of scanning steps for one full 360° B-scan to be 800—this was determined considering the transverse resolution of the OR-PAE imaging mode. Thus, each B-scan image consisted of 400 × 800-pixel values. Due to the aforementioned geometry-dependent nature of EMI noise, while we used the same experimental setup, B-scan images were acquired with different noise levels in terms of the noise amplitude over a number of experimental dates. Thus, among the numerous in vivo datasets we collected, we selected 1000 OR-PAE B-scan images that were least affected by EMI noise.

### 2.2. Data Preparation

For each B-scan image, we performed the Hilbert transform along the radial (or depth) direction for envelope detection and extracted only from the upper region a size of 304 × 800 that contained the most information. Figure 1b shows a typical Hilbert-transformed image. We created the ground truth or target dataset from the acquired 1000 B-scan images using the following semi-manual denoising procedure: First, the intestine region was manually segmented. In other words, this was performed by hand based on our expertise, not by any existing segmentation algorithms, because we wanted to avoid any possible bias when we prepare the training data. Second, while the intestine region remained intact, we removed the noise outside of the region by thresholding. That is, if a pixel value was higher than a threshold value, we assigned the minimum of the adjacent pixel values to the pixel. Indeed, this threshold value is selected as twice as large as the thermal noise level. The whole preprocessing procedure is the same as in reference [29], where all the details can be found.

After the ground-truth dataset was prepared, we added random EMI noise (i.e., noise streaks) to some B-scan images in the ground-truth dataset to prepare a noisy input dataset.

We randomly divided the 1000 ground-truth images into two groups: one group of 700 images for training and the other group of 300 images for validation. To prevent overfitting, we applied random translations in both horizontal and vertical directions during the training. For a test dataset, we prepared 200 images from rat colonoscopy data and added to these images between 100 to 400 noise streaks at random, using the process described above.

### 2.3. CNN Architectures

For signal detection in the noisy images, four types of neural network architecture were implemented: U-Net, Segnet, FCN-16s and FCN-8s, as shown in Figure 1c. These architectures are traditionally used for semantic segmentation. They were considered suitable for our application because EMI noise has structural characteristics that are distinctive from the usual structures in PAE images. We believed that denoising methods based on the idea of semantic segmentation should be able to separate locally connected vertical patterns from the rest of the images and also minimize the influence of the denoising process on noise-free pixels. For this purpose, an output regression layer was used to substitute for the output pixel classification layer, and the softmax layer was removed.

U-Net is a CNN model, based on the fully convolutional network (FCN), which has proven itself able to achieve high accuracy in biomedical image segmentation [65]. It has encoder-decoder architecture, in which an image is first contracted to extract feature maps and then expanded back to its original size [65]. In detail, an image of size 304 × 800 × 1 undergoes a convolution process with 643 × 3 convolutional filters twice to create a feature map of size 304 × 800 × 64. After each convolution, ReLU is applied [66] to set the negative values to 0. Next, the max-pooling layer [67] of 2 × 2 is applied to decrease the image size from 304 × 800 to 152 × 400, with the feature channel doubled (i.e., the feature map of size 304 × 800 × 64 becomes size 152 × 400 × 128). This process continues until the image dimension is reduced to 19 × 50 × 1024. After the image size shrinkage, further dropout layers are applied to decrease the third dimension for the feature channel, and thus decrease the complexity of the neural network [68]. To bring the feature maps back to the original size, the up-sampling process is used. The shrunk input undergoes a series of depth concatenations with features maps from the encoder of the convolution and ReLU activation layers. The increase in the image size is performed by transposed convolution [69].

Similar to U-Net, Segnet is also an encoder-decoder, which is a small easy-to-train network invented for scene understanding [70]. However, unlike U-Net, it does not use depth concatenation, but uses max-pooling indices for the max-unpooling image restoration process. Moreover, Segnet applies a batch normalization layer [71] after convolution to normalize the input.

FCN-16s and FCN-8s are fully convolutional networks (FCN) that provide finer segmentation and regression of images. Indeed, FCN combines features from the deep rough layers and the shallow detailed layers to make the output results finer [72]. Unlike Segnet and U-Net, FCN-16s apply upsampling from the last two max-pooling layers and fusion, while FCN-8s apply upsampling from the last three layers with further fusion [72]. Before the fusion, transposed convolution is performed. More detailed structures of the applied four networks are provided in Appendix A.

### 2.4. CNN Training and Hyperparameter Tuning

We employed L2 regression loss based on the half mean squared error for the training of the CNNs. Each of the CNN weights, *θ*, was updated during the training process based on the following loss function:(1)$$L\left(\theta \right)=\frac{1}{K}\sum_{p=1}^{K}\left(\frac{1}{2},\sum_{i=1}^{H},\sum_{j=1}^{W},{\left(y_{p},\left(i,,,j\right),-,{x\left(\theta \right)}_{p},\left(i,,,j\right)\right)}^{2}\right)+\lambda {\left\|\theta \right\|}^{2}$$
where *x(θ)_p_* is a denoised network output image, *y_p_* is a ground-truth target image, *H* and *W* are the image dimensions, *K* is the total number of images in the mini-batch, and *λ* is a regularization parameter. Please note that *y_p_*(*i*,*j*) is the value of the image *y_p_* at the pixel (*i,j*). To confirm training progress, we used the root-mean-squared error (RMSE) as shown below:(2)$$\mathrm{RMSE}\left(y_{p},,,{x\left(\theta \right)}_{p}\right)=\sqrt{\frac{1}{H*W}\sum_{i=1}^{H}\sum_{j=1}^{W}{\left(y_{p},\left(i,,,j\right),-,{x\left(\theta \right)}_{p},\left(i,,,j\right)\right)}^{2}}$$

RMSE computes pixel-wise error after the noise removal process from a trained CNN on each occasion. For the network parameter update, we employed the Adam optimizer [73]. The learning rate drop factor and period were 0.3 and 10, respectively, and the mini-batch size was 1. Validation RMSE and loss were checked every 50 iterations of the network parameter update. Hyper-parameters, that is, CNN’s initial learning rate, number of epochs, and parameters for L2 regularization were tuned, based on the Bayesian hyper-parameter optimization method, which uses the Bayes theorem and the Gaussian process to estimate the best model hyperparameters [74] and to save time and memory from exhaustive parameter space sweeping. For each network architecture, the best hyperparameters were chosen with the lowest RMSE from the validation set after 10 iterations. All the implementation and training experiments were performed using the experiment Manager Toolbox in MATLAB 2021a (MATLAB) on the PC with Intel(R) Core (TM) i7-10700 CPU @ 2.90GHz, RAM 64.0 GB, and Nvidia RTX 3090 24 GB GPU. Specific hyper-parameters used in the experiments are shown in Table 1.

## 3. Results

### 3.1. Performance Comparison of Trained CNN Architectures

In Figure 2, the RMSE during training is shown for each architecture with the training and validation sets. 

We observed that the terminal value and the speed of convergence of RMSE from the validation set varied among the CNN architectures. In the case of Segnet, the initial and terminal RMSE was not much changed, which means there was no progress in training. By comparison, the FCN-16s, FCN-8s, and U-Net architectures all show rapid decreases in RMSE during the first few steps of training; however, the FCN architectures resulted in no significant further improvements. The architecture that converged to the lowest RMSE from the training record is U-Net. We also implemented the structural similarity index measure (SSIM) [75] to verify quantitatively that the observation target signal remained as intended:(3)$$0\leq \mathrm{SSIM}\left(x,,,y\right)=\frac{\left(2,\mu_{x},\mu_{y},+,c_{1}\right)\left(2,\sigma_{xy},+,c_{2}\right)}{\left(\mu_{x}^{2},+,\mu_{y}^{2},+,c_{1}\right)\left(\sigma_{x}^{2},+,\sigma_{y}^{2},+,c_{2}\right)}\leq 1$$where *x* and *y* are noise-removed and ground-truth images; *μ_x_* and *μ_y_* are the means of *x* and *y*, respectively; *σ_x_* and *σ_y_* are the standard deviations of *x* and *y*, respectively; *σ_xy_* is the cross-correlation of *x* and *y*; and *c_1_* and *c*_2_ are the variables used to stabilize the division with a small denominator. Similar to RMSE, there is another well-known measure called the mean absolute error (MAE):(4)$$\mathrm{MAE}\left(y_{p},,,{x\left(\theta \right)}_{p}\right)=\frac{1}{H*W}\sum_{i=1}^{H}\sum_{j=1}^{W}\left|y_{p},\left(i,,,j\right),-,{x\left(\theta \right)}_{p},\left(i,,,j\right)\right|$$that we computed for fair comparison.

In Figure 3, a summary of noise removal performance by the trained CNN architectures is shown. The results distinctively indicate that the U-Net structure outperformed the other architectures in terms of RMSE, SSIM, and MAE. Although all four architectures belong to fully convolutional neural networks, the slight differences in detail between the architectures resulted in clear distinctions in the different performance scores generated. The key difference among the network structures is the information that is used in the restoration process. To explain this in detail: Segnet uses indices from max-pooling for max unpooling between the same depths for restoration, whereas FCN-16s and FCN-8s utilize encoded information at the last two (or three) max-pooling layers, which are inadequate to restore the observation target signals. In contrast, U-Net utilizes concatenation through the skip connections between the same depth levels, which compensates for the information lost at the max-pooling layers in the restoration of signals at the decoding part. The consequences of these architectural differences are well represented in Figure 4. Regardless of the noise level, all the trained CNNs were able to remove noise from the background region. However, the quality of the various signal restorations turned out to be significantly different. In fact, Segnet failed to recover most of the observation target signals, whereas the FCN structures restored blurry signals. In contrast, U-Net restored the observation target signals with little damage.

In comparison with the results by deep learning-based methods shown in Figure 3 and Figure 4, we also tested a few well-known classical filtering methods for the noise removal and present related results in Appendix A. Due to the distinctive nature of the EMI noise, which the classical methods had not been modeled for, the superiority of the deep-learning approaches to the classical ones is clearly observed for the EMI noise removal. This confirms the feasibility of deep learning-based EMI noise removal, which is the main objective of this study.

### 3.2. Performance Test for New In Vivo Data

As presented in the previous section, the U-Net architecture exhibited the best performance for image-to-image regression tasks in terms of the RMSE, SSIM and MAE. Moreover, it did not exhibit any notable performance degradation, even when the added noise density was increased. Thus, to demonstrate the noise removal capability of the established AI algorithms, we chose the U-Net-based algorithm and assigned it to a task to remove EMI noise from two new rat colorectum OR-PAE in vivo datasets at different levels in terms of amplitude and density.

Due to the non-availability of the related method for quantitatively defining the noise levels, we cannot present the related values at present. However, as shown in Figure 5a, which presents radial-maximum amplitude projection (RMAP) images, the U-Net algorithm removed EMI noise from the two in vivo datasets to a fairly satisfactory level, and thus, the finest, mesh-like capillary networks, which typically have a hole size of ~50 µm, were able to appear more clearly in the magnified denoised images. The RMAP images presented were produced using their corresponding C-scan datasets, which consisted of 3000 B-scan image slices, and the AI algorithm performed the noise removal task in a B-scan image state for all the image slices involved. To show the related process, in Figure 5b, we present two sets of before and after B-scan images, which were taken from the lines marked with dashes and included in the RMAP images presented in Figure 5a.

To show the effect of the EMI noise removal from the two in vivo datasets more clearly, we plotted volume-rendered images for the four RMAP images presented in Figure 5a and present the results in Figure 6. As shown, the vascular structures that we were looking for through the colorectum imaging experiments are more clearly visible in the denoised (after) images, whereas those are hardly visible in the raw images (before) because they were superimposed with EMI noise that appeared like countless thorns around the vasculatures. For reference, several dark regions in the denoised RMAP images in Figure 5a appeared dark because corresponding colorectal wall portions were imaged outside the working distance of the endoscopic probe rather than because related data values were lost during the denoising process.

## 4. Discussion

In this study, we investigated CNN-based EMI noise removal algorithms, U-Net, Segnet, FCN-16s, and FCN-8, which have been widely used in biomedical image processing for image-to-image regression. We evaluated their performance and effect in terms of reconstruction quality assessments such as RMSE, SSIM and MAE. Although these CNNs have already been applied to previous PA image segmentation and regression-related research [40,41,42,43,44], our work is the first to apply the architectures to remove EMI noise from PAE images, and more specifically from OR-PAE images. Although CNN-based algorithms have shown outstanding performance in white Gaussian noise removal [76] or in impulse noise removal [77], it was not known whether they have the ability to remove other types of noise, especially a type of noise as peculiar as EMI noise. As we have observed, the denoising performance by U-Net was satisfactory, with most of the EMI noise present in the new test image dataset (i.e., Figure 5) being automatically removed, requiring neither additional laborious manual pre-processing nor image segmentation, thereby achieving computational efficiency. Moreover, it seemed that the intrinsic nature of the U-Net architecture kept the EMI noise-free regions as little affected by the denoising process as possible, minimizing data loss in important signal areas, such as the intestine wall, as the healed RMAP images enable us to see the tiny blood vessel mesh structure more clearly, with little EMI noise.

Looking back at the history of science, there are many instances where early work in a discipline was conducted or achieved by imperfect, even primitive, recently constructed or invented systems or instruments. PAE is no exception, and it has been following a similar process. Although the first conceptual proposal was reported about 20 years ago [18], and there was an expectation that it could make an important clinical contribution [9], its progress has been very slow, as is evident from the fact that in vivo imaging of the gastrointestinal system of animals is not routinely performed in related research. Apart from its detailed discussion of the underlying technical challenges, the main contribution of this study is to shed light on the possibility of AI-based algorithms for EMI noise removal and the provision of promising computationally efficient EMI-noise removal approaches in newly built PAE systems, both which were presented clearly in the context of this paper.

Speaking about our own research from this perspective, before developing this kind of noise removal algorithm, we spent about a year on the construction of our first PAE system, but we acquired unsatisfactory image data that had a noticeable amount of the EMI noise. This made it difficult for us to accurately identify what kinds of morphological features were included in the acquired vascular images, although we could be sure of the presence of blood vessel-like structures in the PA-RMAP images acquired from a rat colorectum. Due to the poor performance, we undertook rebuilding of the related endoscopic probe and amplifier circuit. However, no matter how many times we repeated the imaging procedure, we ended up with similar unsatisfactory results without realizing the key factors. Consequently, we considered a change in our original plan of creating an OR-PAE system to creating an AR-PAE system as a way to take a step backward. This change was never made, which was very fortunate, because we found interesting vascular structures, such as honeycomb-like capillary networks and hierarchically varying larger vascular networks, in the raw PA-RMAP images after spending a month removing EMI noise-affected pixels from several thousands of B-scan images one-by-one, manually. This experience motivated us to go forward with the development of an efficient EMI noise removal algorithm. Although the current algorithm investigated in this study is not 100% perfect, we hope that the EMI noise is no longer a hindrance when working on PAE in the newly applied areas.

As an alternative to an AI-based denoising approach, one may consider the use of already-existing well-developed classical denoising methods for the removal of EMI noise. However, as we previously mentioned in the introduction section, we came to the conclusion in our literature search that no prior report has addressed the problem of the removal of EMI noise that appeared as the rain-like striking pattern, as in our case. Thus, in our previous study [29], we developed a dedicated algorithm that could detect EMI noise-affected pixels based on the calculation and comparison of a signal gradient to its adjacent pixels along the transverse direction, which eventually turned out to be not as satisfactory as the current AI-based one. Again, the classical methods were unable to correctly distinguish an EMI noise-affected pixel from a normal capillary signal because the two patterns become similar, as the transverse resolution of the developed PAE probe was at the OR level. Thus, we had to apply the algorithm only to the area where there was no intestinal signal after first performing a manual image segmentation process on the B-scan images that were affected. 

That is, our point is that although it would not be difficult to remove the EMI noise in AR-PAE images by applying a non-AI-based approach, it is not so simple if the problem is related to OR-PAE images. Of course, in the case of OR-PAM [10,14], although it shares the same technical basis as OR-PAE, the EMI noise removal issue has not been a major concern because it is relatively easy to build a perfect hardware system that is not affected by EMI noise. On the other hand, we expect that an AI-based noise removal issue, such as that addressed in this study, could also be important in OR-PAM research because constructing a laser diode or an LED-based PAM system is emerging as a new important subject of PAT these days, while the SNR of related systems is known not to be high enough due to the relatively lower power of the light sources [13,78,79,80].

Although the feasibility of a deep-learning-based EMI noise removal strategy has been successfully demonstrated in this study, there are several limitations. First, we trained the AI algorithms using only OR-PAE images acquired from the urethra and colon, and while such an endoscopic device could also be utilized for other organs, such as the blood vessels, esophagus, and bile ducts, limitations may have been introduced by training the AI algorithms to correctly recognize only the EMI noise features, and not to be affected by the morphological difference between the colorectum and the urethra. Therefore, in future studies, more varied datasets should be acquired and included in the training process. Second, while our endoscopic system acquired not only the PA images but also the US images, we did not consider the removal of the EMI noise from the US images. Although this issue seems relatively easy, we would like to consider this research for completion. Lastly, image segmentation and regression of the ground-truth creation algorithms were dependent on a manual boundary detection, requiring expertise in PAE imaging, and taking an excessive amount of time. This time-consuming and laborious work to achieve clean PAE images should be taken into account within the deep-learning-based algorithm suggested. Furthermore, the optimized noise removal CNN model could be embedded into the PAE system for real-time denoising visualization. We think that these tasks require a more comprehensive investigation of different types of observation objects and more training datasets for the greater generalization of the CNN models, for which we have laid a foundation in this study.

## 5. Conclusions

In this study, we investigated the feasibility of deep-learning-based EMI noise removal in relation to an OR-PAE image processing using four CNN architectures (U-Net, Segnet, FCN-16s and FCN-8s). We obtained satisfactory results from the U-Net-based architecture, removing the EMI noise from 2D images and visualizing a clearer 3D structure of mesh-like capillary networks with a hole size of ~50 µm included in a test of in vivo data. That is, our work has made EMI noise removal a tractable problem via deep learning. As the developed algorithms were trained using the OR-PAE image data acquired from rat colorectums and a rabbit urethra by using 40 MHz US transducers, we expect that the algorithms achieved can be used to remove the EMI noise included in other OR-PAE images based on similar device specifications. To make the machine learning architecture a better fit for the EMI noise removal, we think that classical non-AI approaches should also be taken into consideration in future research for the characterization of the noise features from both the engineering and mathematical points of view. We believe that such information in the AI framework will definitely enhance the final denoising outcomes and have a wider range of applications in PAE imaging.

## Figures and Tables

**Figure 1 sensors-22-03961-f001:**
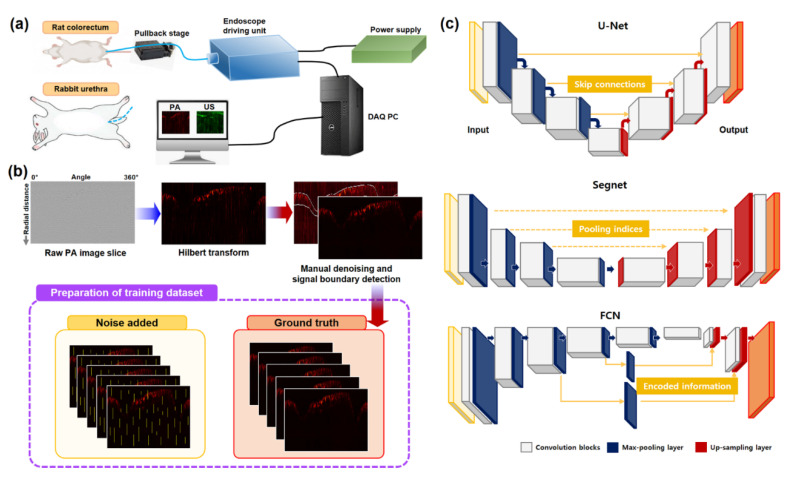
Overview of training data preparation and CNN architectures considered in this study: (**a**) The imaging system we set up, (**b**) data preparation procedure. The EMI noise pattern looks very similar to the blood vessels in the Hilbert-transformed image. (**c**) CNN architectures: U-Net, Segnet, and FCN-16s. Further details of the networks are provided in Appendix A.

**Figure 2 sensors-22-03961-f002:**
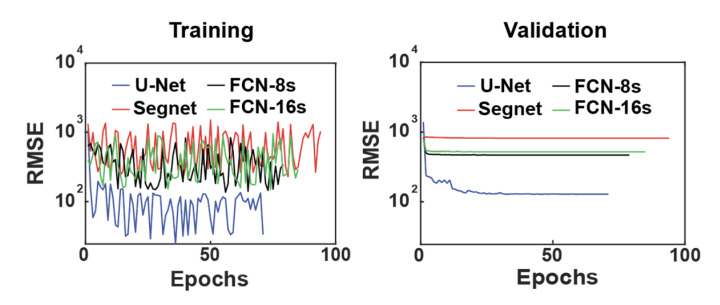
RMSE during training.

**Figure 3 sensors-22-03961-f003:**
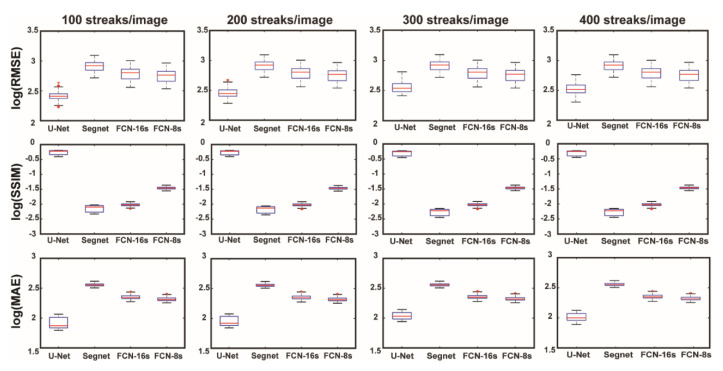
Performance metric comparison between trained CNNs’ log-scaled RMSE (top) and SSIM (middle) and MAE (bottom) for each noise level (streaks per image) using the test set. The smaller the RMSE and MAE and the larger the SSIM, the better reconstruction we have. Further comparison with classical approaches can be found in Appendix A.

**Figure 4 sensors-22-03961-f004:**
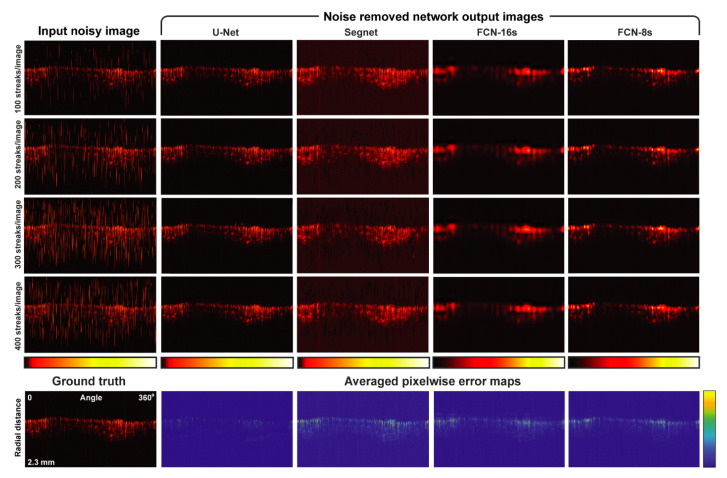
Visualized noise removal example for a single B-scan image with different noise levels. The noise removal performance of each architecture is shown as a pixelwise error map, which calculates the difference between the ground truth and averaged network outputs from all tested noise levels. Further comparison with classical approaches can be found in Appendix A.

**Figure 5 sensors-22-03961-f005:**
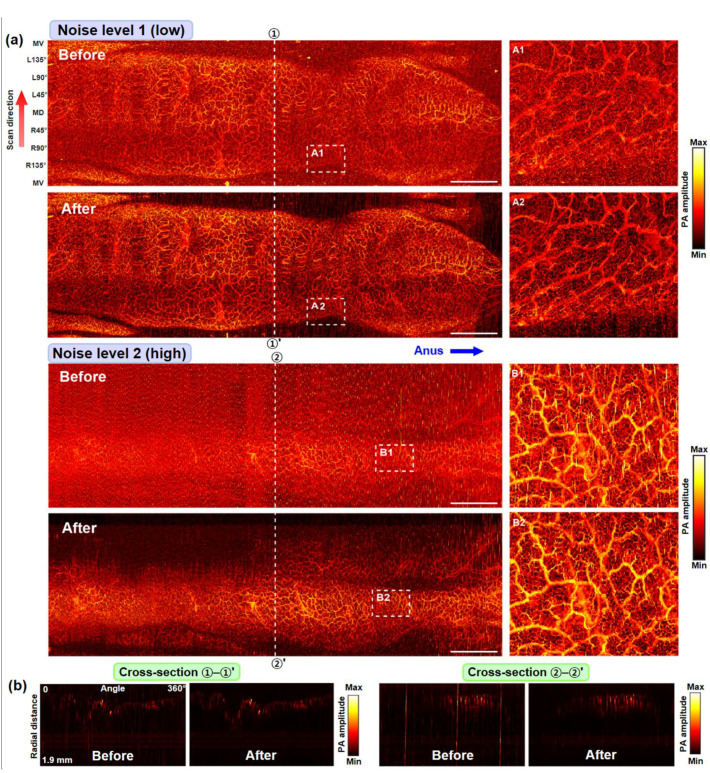
Performance test results for two in vivo rat colorectum test datasets with different EMI noise levels: (**a**) whole PA-RMAP images (left) and magnified images for the dashed box regions (right). MD, mid-dorsal; MV, mid-ventral; L, left; R, right. Scale bars, 5 mm (horizontal only). (**b**) B-scan (or cross-sectional) images for the marked positions in (**a**).

**Figure 6 sensors-22-03961-f006:**
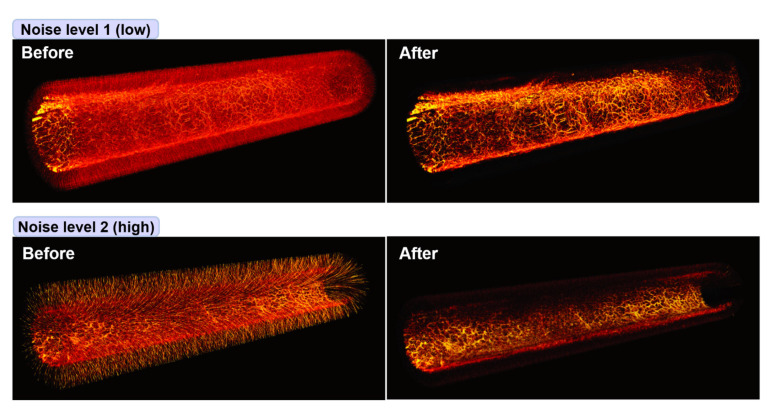
Three-dimensional rendering of the two in vivo rat colorectum test datasets presented in Figure 5. Left and right images correspond to before and after denoising, respectively. Each image corresponds to a range of over ~5 cm with an image diameter of ~7 mm.

**Table 1 sensors-22-03961-t001:** Summary of other parameters for the networks.

	Initial Learning Rate	Epoch Number	L2 Regularization	Training RMSE
U-Net	0.0002	70	0.0465	33.2993
Segnet	0.0010	93	0.0497	1028.6909
FCN-16s	0.0003	84	0.0166	291.4638
FCN-8s	0.0002	78	0.0205	344.4395

## Data Availability

Data available on request from the authors.

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
