# Peer review of "Deep-Learning-Based Algorithm for the Removal of Electromagnetic Interference Noise in Photoacoustic Endoscopic Image Processing"

_sensors, 2022, doi:10.3390/s22103961_

Round 1

Reviewer 1 Report

Author entitled “Deep Learning-Based Algorithm for the Removal of Electro- 2 magnetic Interference Noise in Photoacoustic Endoscopic Image Processing” . Author reported the CNN-based EMI noise removal algorithms. Here I would like point out few corrections to be done.

  1. Final outcome of the proposed model could be addressed in the abstract as well as conclusion section.
  2. Any manuscript survey is more important to discuss about the existing models and its results. Here author discussed few information in the introduction section. Better authors need to include literature survey as separate section or Author can discuss in the table format of the existing works with the final results achieved in the existing models.
  3. In section.2 author explained about the data preparation. Its good but how these data are processed? What is the techniques? all these information’s are missing in the manuscript. Since pre-processing and segmentations are very important in any image processing techniques.
  4. Author did not represent the mathematical equations as number. It should be represented by equation number as well as some parameters are missing in the explanation line no 215 to 220. Author must describe all the parameters were used in the mathematical derivations.
  5. Author focused on noise removal areas with all the relevant techniques. Its appreciable but finally what’s the methodology achieving good results in terms of accuracy level. Author could include additional visualization representation for representing accuracy compare to existing models.

Thank you

Reviewer 2 Report

Deep learning was used in this paper to filter noise in photoacoustic images. Two metrics were employed: the root mean squared error (RMSE) and the structural similarity index (SSIM). The results appear positive, but the following changes should be made before publication:

- The limitations of each conventional technique should be explicitly highlighted in the introduction section.

- To clarify the stance of this work further, the authors could underline the differences between various methodologies.

- The introduction must address the broad variety of applications.

- A summary of the contributions should be presented at the end of the introduction section.

- In the introduction, the findings of the present research work should be compared with the recent work in the same field in order to claim the contribution made. Kindly provide several references to substantiate the claim made in the abstract (that is, provide references to other groups who do or have done research in this area).

- Related research and literature review sections should be included to provide a brief overview of related noise removal and filter studies, including methodology and applications. Authors can examine some of the most recent related publications from prestigious journals such as IEEE/ACM Transactions, Elsevier, Inderscience, Springer, Taylor & Francis, and others.

- Why are only RSME and SSIM utilized for performance evaluation? It is suggested that the authors use well-known performance measurements such as mean absolute error (MAE) and normalized color difference (NCD) to measure the performance of noise filtering and removal.

- The optimized hyperparameters should be revealed and explained in the paper.

- Because Figure 1 is too brief, the deep learning architecture used should be presented and explained to increase the study's reproducibility.

Reviewer 3 Report

In this paper, the authors apply the deep learning-based algorithms involving the U-Net, SegNet, and FCN to remove the EMI noise from the PAE B-scan images.  This work is interesting and the results verify the superiority of the deep learning model. The overall structure is clear and the writing quality of the manuscript still has the space to be improved. The following are my comments for this paper. 

  1. The authors compare the performance of four different deep learning models over the image noise removal task. In the discussion section, the authors mention they have used the classic denoise method before but the results are not good. I suggest the authors add the results of the classic method in figure 3 and figure 4, which is clear for the readers to know how deep learning-based methods outperform the classic method.
  2. For the loss function mentioned in section 2.4, I assume the authors added the L2 regression to the original RSME to solve the overfitting problem. For the first equation in section 2.4, I don't understand the meaning of "1/2", does it mean the square root of RSME? If so, I think the position of "1/2" is wrong. Please double-check it.
  3. For figure 2, we can clearly see the training loss curve fluctuates while the loss curve for the validation is smooth. Do the authors know the reason for it? I notice the batch size of training is 1, so the small batch size may be one possible reason for it.
  4. I notice the original image size is 304*800 and after the encoder, the image size is 19*25. I assume the actual value should be 19*50.  Can author double check this value?
  5. In section 2.4, the authors apply the Bayes method for the hyper-parameter tuning of four deep learning models, the authors should list the hyper-parameters for each model in the paper.

Round 2

Reviewer 1 Report

My previous comments was addressed by the author. Thank you.

Reviewer 2 Report

The authors have addressed the majority of the comments, and the current format appears adequate.

Reviewer 3 Report

The authors address all my comments in this revised manuscript and I recommend it can be accepted.